# Food Addiction among Female Patients Seeking Treatment for an Eating Disorder: Prevalence and Associated Factors

**DOI:** 10.3390/nu12061897

**Published:** 2020-06-26

**Authors:** Marie Fauconnier, Morgane Rousselet, Paul Brunault, Elsa Thiabaud, Sylvain Lambert, Bruno Rocher, Gaëlle Challet-Bouju, Marie Grall-Bronnec

**Affiliations:** 1Addictology and Psychiatry Department, Hôpital Saint Jacques, University Hospital of Nantes, 85 rue Saint Jacques, CEDEX 1, 44 093 Nantes, France; marie.fauconnier@chu-nantes.fr (M.F.); morgane.rousselet@chu-nantes.fr (M.R.); elsa.thiabaud@chu-nantes.fr (E.T.); sylvain.lambert@chu-nantes.fr (S.L.); bruno.rocher@chu-nantes.fr (B.R.); gaelle.bouju@chu-nantes.fr (G.C.-B.); 2Inserm, SPHERE U1246 « methodS in Patients-centered outcomes and HEalth ResEarch », Nantes University, Tours University, 22 boulevard Benoni Goullin, 44 200 Nantes, France; 3Department of Psychiatry & Addiction, University Hospital of Tours, 2 boulevard Tonnellé, 37 000 Tours, France; paul.brunault@univ-tours.fr; 4Inserm UMR 1253, iBrain, Tours University, 10 boulevard Tonnellé, 37 000 Tours, France; 5Qualipsy EE 1901, Tours University, 3 rue des Tanneurs, CEDEX 1, 37041 Tours, France

**Keywords:** food addiction, eating disorder, anorexia nervosa, bulimia nervosa, binge eating disorder, YFAS, addictive disorder, eating addiction, addictive-like eating behavior

## Abstract

The concept of “food addiction” (FA) has aroused much focus because of evidence for similarities between overeating and substance use disorders (SUDs). However, few studies have explored this concept among the broad spectrum of eating disorders (ED), especially in anorexia nervosa (AN). This study aimed to assess FA prevalence in ED female patients and to determine its associated factors. We recruited a total of 195 adult women with EDs from an ED treatment center. The prevalence of FA diagnosis (Yale Food Addiction Scale) in the whole ED sample was 83.6%; AN restrictive type (AN-R), 61.5%; AN binge-eating/purging type (AN-BP), 87.9%; bulimia nervosa (BN), 97.6%; and binge-eating disorder (BED), 93.3%. The most frequently met criteria of FA were “clinically significant impairment or distress in relation to food”, “craving” and “persistent desire or repeated unsuccessful attempts to cut down”. An FA diagnosis was independently associated with three variables: presence of recurrent episodes of binge eating, ED severity, and lower interoceptive awareness. In showing an overlap between ED and FA, this study allows for considering EDs, and AN-R in particular, from an “addictive point of view”, and thus for designing therapeutic management that draws from those proposed for addictive disorders.

## 1. Introduction

Similarities between overeating and substance use disorder (SUD) were envisaged decades ago. In 1956, Theron Randolph mentioned for the first time the term food addiction (FA), with the hypothesis that certain food, as psychoactive substances, produces a “common pattern of symptoms descriptively similar to those of addictive processes” [1]. Subsequently, many studies have found similarities between certain forms of overeating and SUDs, especially studies conducted with animal models, notably rat models, in which the overconsumption of sweet food led to specific behavioral modifications (bingeing, withdrawal and cross-sensitization) [2,3] and neurochemical signs were also observed in models of substance dependence [2,3,4]. In humans, neuroimaging studies, notably those conducted with obese patients with FA, have also suggested the involvement of brain dopamine (DA) pathways and reward circuitry, and similarities with substance dependence have been observed as well [3,5,6].

The increasing prevalence of obesity, reflecting multiple factors that include the overall easy access to highly palatable energy-dense foods, linked with the food industry’s efforts to boost sales, has contributed to making the concept of FA more popular. In 2009, Gearhardt et al. [7] therefore proposed an operationalization of a measure of FA by extrapolating the diagnostic criteria for substance dependence (Diagnostic and Statistical Manual of Mental Disorders, fourth edition Text Revised: DSM-IV-TR) [8] to hyperpalatable foods (i.e., foods high in fat and/or sugar). These criteria included (1) tolerance, (2) withdrawal, (3) consumption of larger amounts or over a longer period than was intended, (4) loss of control, (5) a great deal of time spent, (6) important activities are given up or reduced, and (7) persistent use despite damage. As in SUD, the presence of three (or more) of the criteria, as well as a clinically significant impairment or distress, have been suggested as necessary to characterize FA. This has led to the validation of a new evaluation tool, the Yale Food Addiction Scale (YFAS), which is a self-administered questionnaire assessing eating behavior in the past 12 months, with 25 questions exploring the 7 DSM-IV-TR extrapolated criteria. This tool has shown good internal consistency (Kuder–Richardson α = 0.86), good convergence with measures of similar constructs (i.e., binge eating, emotional eating), good construct validity relative to dissimilar constructs (i.e., alcohol use, impulsivity), and good incremental validity toward binge-eating behavior and has been translated in several languages [9]. After publication of the DSM-5, a new version was developed in 2016, the YFAS 2.0, allowing a more dimensional approach with the exploration of the 11 DSM-5 criteria of SUD as applied to food through 35 questions [10].

The prevalence of FA, determined by the YFAS, varies greatly across samples, ranging from 0 to 25% in nonclinical samples [11,12], from 14 to 57.8% in prebariatric surgery samples [11,13], and from 70 to 90% in samples of patients suffering from eating disorders (ED), especially bulimia nervosa (BN) and binge eating disorder (BED) [11]. In most studies, YFAS symptoms were positively associated with BMI scores, and elevated YFAS scores have been observed in patients suffering from obesity [11,14]. Moreover, FA is associated with clinical characteristics that are commonly found with other addictive disorders: depression, anxiety [11], comorbid addictive disorders [15], posttraumatic stress disorders [14] and ADHD [16,17]. People with FA show more insecure attachment styles [18] and higher impulsivity [17,19,20], a classical trait of addictive disorders. In ED samples, FA has been associated with a more severe eating pathology and psychopathology, such as higher negative urgency, higher reward dependence and higher harm avoidance [20,21].

To date, the YFAS has mainly been used in overweight or obese patients (with or without BED) or in patients suffering from BN. The common factor among all these patients is overeating. To our knowledge, studies examining the links between anorexia nervosa (AN), a disorder characterized by restriction in energy intake, and FA are rare. Only two studies assessed FA prevalence in a sample of ED patients that included AN patients [20,21]. Nevertheless, AN is considered a counterpart of BN, and though FA and EDs of all types differ, they also show several similarities, as shown in Table 1. The concept of FA remains widely debated, and some authors argue that eating rather than food is addictive, underlining the behavioral dimension of this addiction. In light of these issues, we conducted a study with a sample of patients suffering from ED, characterized by eating behavioral symptoms ranging from those of AN (restricting: AN-R or binge-eating/purging: AN-BP types) to those of BN and BED.

We aimed to estimate (i) the prevalence of FA among ED patients in general and according to the type of ED. We also aimed (ii) to assess the most commonly fulfilled criteria of the YFAS among ED in general and according to the type of ED and (iii) to determine the clinical and psychopathological correlates of FA in ED patients in general with an explorative approach, including the assessment of characteristics usually associated with addictive disorders and particularly with FA. It was first hypothesized that the prevalence of FA among ED patients would be important because of the similarities between those two disorders. Second, no clear hypothesis was made concerning the most fulfilled YFAS criteria that would be found in ED, except that the two physical criteria (tolerance and withdrawal) would not be very prevalent. Third, regarding the literature cited above, we hypothesize that the presence of FA would be associated with ED severity and the binge-eating episodes, and we also expected that FA would be associated with more comorbid addictive disorders, ADHD in childhood and trauma history, higher impulsivity, greater reward dependence, harm avoidance, and more insecure attachment styles.

## 2. Methods

### 2.1. Procedure and Ethics

Our Addictology and Psychiatry Department, based in the University Hospital of Nantes, France, is especially specialized in ED management (i.e., AN, BN, and BED) and is recognized as a National Reference Center in France. To receive treatment in our ED unit, patients must be referred to us by a medical professional. We provide physical, psychological and social care in accordance with the guidelines for ED management [22,23,24]. The care objectives of our unit are as follows: (i) to restore patients to a healthy weight, (ii) to alter core dysfunctional symptoms and attitudes related to ED (excessive concerns about body shape and weight, dietary restriction, purge and binge symptoms, etc.), and (iii) to manage all other negative features associated with ED (anxiety and depressive symptoms, low self-esteem, etc.) Treatment is primarily conducted in an outpatient format, with inpatient treatment provided only if necessary. Treatment is adapted to patient heterogeneity and often differs from one patient to another, in accordance with ED treatment guidelines.

Since September 2012, an in-depth clinical assessment has been systematically carried out for all new ED patients referred to our unit for treatment. The aforementioned assessment, which is part of the EVALuation of behavioral ADDictions (EVALADD) cohort (NCT01248767), occurs prior to the first medical consultation (at inclusion) and is then readministered at predefined intervals (at 6 months, at 12 months, and then every year). This assessment aims to highlight the risk factors involved in ED initiation and persistence. The main criteria for inclusion in the cohort were as follows: age of 15 years or older and a diagnosis of ED as defined by the DSM. Patients with cognitive impairment or difficulties reading or writing French were not included. All patients participated in a face-to-face semistructured interview and completed self-report questionnaires (see Section 2.3). Qualified and experienced staff members performed these assessments. Inclusion in the EVALADD cohort is still in progress.

The EVALADD cohort study is conducted in accordance with Good Clinical Practice Guidelines and the Declaration of Helsinki, with approval from the local ethics committee (Groupe Nantais d’Ethique dans le Domaine de la Santé, GNEDS, Nantes—Number 6 September 2012). All participants provide written informed consent, including consent from parents or guardians for participants under age 18. No compensation is given for participation.

For this specific study, we only used data collected at inclusion.

### 2.2. Participants

The participants were patients from the EVALADD cohort. For the present study, the specific inclusion criteria were as follows: (i) having a current diagnosis of AN (AN-R or AN-BP), BN or BED according to the DSM at inclusion; (ii) being included in the EVALADD cohort; and (iii) being a woman.

A total of 195 patients were included in this study. Sixty-five patients (33.3%) were diagnosed with AN-R, 33 (16.9%) with AN-BP, 82 (42.1%) with BN and 15 (7.7%) with BED. The flow chart of patient selection is presented in Figure 1.

### 2.3. Measures

#### 2.3.1. Sociodemographic Characteristics

Sociodemographic data included age and gender.

#### 2.3.2. Eating Disorder Characteristics

• *Type of ED*

AN and BN diagnoses were made according to the ED sections of the fifth version of the Mini International Neuropsychiatric Interview (MINI). It is a structured diagnostic interview that enables rapid and systematic investigations of the main axis 1 psychiatric disorders, according to DSM-IV criteria [25,26]. From 2017, we used an adapted version of the MINI to take into account the revised diagnostic criteria of the DSM-5 [27]. Questions were added to diagnose BED according to DSM-IV (and then DSM-5) criteria. Age at ED onset and disease duration were also collected.

• *Severity of ED*

The Morgan–Russell Outcome Assessment Schedule (MROAS) is a structured interview that covers various clinical symptoms of ED and their repercussions on patient functioning in the past six months [28]. The questionnaire consists of five subscales exploring food intake and nutritional status, menstrual function, mental state, psychosexual adjustment, and socioeconomic status. Each subscale was scored from 1 to 12, with a higher score indicating a better outcome in the corresponding field. The average of these five scores was used as the MROAS total score, with potential results ranging from 1 to 12.

• *Characteristics of ED*

The Eating Disorder Inventory-2 (EDI-2) is a 91-item self-assessment questionnaire that evaluates the symptomatology and behavior associated with ED [29]. It examines 11 dimensions: “drive for thinness”, “bulimia”, “body dissatisfaction”, “ineffectiveness”, “perfectionism”, “interpersonal distrust”, “interoceptive awareness”, “maturity fears”, “asceticism”, “impulse regulation” and “social insecurity”. Answers are rated on a 6-point Likert-type scale ranging from “never” to “always”. Each of these dimensions can be independently analyzed, and a score was calculated for each item. The internal consistency values for the EDI-2 dimensions are between 0.44 and 0.93.

• *Food addiction*

The Yale Food Addiction Scale (YFAS) was designed to identify those exhibiting addictive-like eating behavior toward certain types of foods high in fat and/or sugar [7]. The YFAS is composed of questions based upon substance dependence criteria in the DSM. The DSM-IV criteria were used for the initial version of the YFAS and showed good convergence with measures of similar constructs (i.e., binge eating, emotional eating), good construct validity relative to dissimilar constructs (i.e., alcohol use, impulsivity), and good incremental validity toward binge-eating behavior. When the fifth edition of the DSM was published, a new version of the YFAS was developed, YFAS 2.0, to take into account the changes made to the substance-related and addictive disorders section and to extrapolate them to food [10]. The YFAS 2.0 version also showed good internal consistency, as well as convergent, discriminant and incremental validity. For the present study, the French version of the initial version of YFAS [30] was used until 31 October 2017, then replaced by the French version of YFAS 2.0 once it was validated [31]. According to Meule and Gearhardt (2019), prevalence rates and correlates of YFAS 2.0 diagnoses are largely similar to those observed with the original YFAS [32].

#### 2.3.3. Other Clinical Characteristics

• *Psychiatric comorbidities*

The fifth version of the MINI (described above) was used. For the purposes of this study, mood disorders (major depressive episode, dysthymia, (hypo)manic episodes), anxiety disorders (panic disorder, agoraphobia, social phobia, obsessive-compulsive disorder, posttraumatic stress disorder, generalized anxiety disorder), psychotic syndrome and SUD (alcohol, psychoactive substances)), current or past, were considered. We also assessed the presence of behavioral addictions with the Minnesota Impulsive Disorders Interview (MIDI) [33,34]. The MIDI is a structured interview that enables rapid and systematic investigations of pathological gambling, hypersexuality, and compulsive buying. In the framework of the EVALADD cohort, we adapted the MIDI to screen other behavioral addictions (videogame, internet, exercise and work addictions). When a behavioral addiction was screened by the MIDI, its diagnosis was confirmed using a specific diagnostic interview. Finally, the Wender Utah Rating Scale-Child (WURS-C) was used to retrospectively screen for childhood attention-deficit/hyperactivity disorder (ADHD) [35,36]. A threshold of 46/100 was defined to identify probable childhood ADHD.

• *Impulsivity*

The French version [37] of the Impulsivity Behavior Scale (UPPS) [38] was used to measure impulsivity. During the data collection, we transitioned to the UPPS-P French short version of the scale [39], which included a fifth new dimension, “positive urgency”. To standardize the results, we reconstructed the four available scores of the new UPPS-P (“negative urgency,” (lack of) “premeditation,” (lack of) “perseverance” and “sensation seeking”) based on the initial UPPS for the first patients.

• *Temperament*

The 125-item version of the Temperament and Character Inventory (TCI-125) is a validated self-report questionnaire. It is used to briefly evaluate 4 temperament dimensions (novelty seeking, harm avoidance, reward dependence and persistence) and 3 character dimensions (self-directedness, cooperativeness and self-transcendence) [40]. For the present study, only temperament dimensions were considered.

• *Attachment*

The Relationship Scales Questionnaire (RS-Q) is a 30-item self-assessment questionnaire that was developed in 1991 [41] and validated in French in 2010 [42]. It is based on the theoretical principles of Bowlby and, more specifically, on the concept of an internal working model to examine four different types of attachments: “secure”, “fearful”, “preoccupied” and “dismissing”. For all items, answers were given on a 5-point Likert-type scale ranging from “not at all like me” to “just like me”. In the French translation study, Cronbach’s alpha coefficient was moderate (α > 0.60), and the intraclass coefficients were good (>0.75).

• *History of Traumatic Events*

We used a revised version of the French Life Events questionnaire (EVE) [43], which was previously used in another study from the EVALADD cohort [44]. The revised EVE questionnaire explores 6 areas (family, professional life, social life, marital and emotional life, health, and other traumatic events). For this study, we focused on the history of physical abuse and sexual abuse.

### 2.4. Outcome Measure

The primary outcome measure was the diagnosis of FA (yes or no) according to the YFAS. When the YFAS version 1.0 was used, FA was diagnosed when 3 or more criteria out of 7 were present during the last 12 months and when clinically significant impairment or distress was endorsed. When the YFAS 2.0 version was used, FA was diagnosed when 2 or more criteria out of 11 were present during the last 12 months and when clinically significant impairment or distress was endorsed.

### 2.5. Statistical Analysis

A descriptive statistical analysis was conducted for the entire sample. Continuous variables are described by means and standard deviations, while categorical variables are presented as numbers and percentages. The prevalence of FA according to YFAS 1.0 and 2.0 was computed for the whole sample and for each ED diagnosis, as well as the frequency of each YFAS criterion for each ED diagnosis. We divided the sample into two groups (“FA” and “No FA”) according to FA diagnosis. Bivariate analyses were conducted to explore the associations between FA and other collected data (sociodemographic, ED and other clinical characteristics). We focused on patients with recurrent episodes of binge eating (at least once a week). We used chi^2^ tests or Fisher’s tests, if necessary, to analyze the categorical variables. For the continuous variables, we used Student’s tests for variables with a normal distribution and Wilcoxon nonparametric tests for variables with a non-Gaussian distribution. For both types of variables (categorical and continuous), differences were statistically significant when the p-value was less than or equal to 0.05.

A multiple logistic regression was performed using an iterative selection procedure to identify the variables that were significantly associated with FA, as assessed by the likelihood ratio test. Variables were entered as candidates for the model if they were associated with the presence of FA in the bivariate analysis with a *p* < 0.20 [45]. Then, nonsignificant variables were removed one at a time starting with the least significant variable (backward procedure), to retain only the variables that provided significant information to the model (*p* < 0.05) [46]. The corresponding odds ratios (OR) and associated 95% confidence intervals were estimated. Discrimination of the final logistic model, which describes the model’s ability to differentiate between the presence and absence of FA, was assessed using the area under the receiver operating characteristic (ROC) curve, and the goodness-of-fit of the model was assessed using the Hosmer–Lemeshow test. The statistical analysis was carried out with TIBCO Statistica^®^ 13.3.0 (Statsoft, Inc. 2300 East 14th Street. Tulsa, OK 74104, USA.). The conditions of validity were verified for all tests and the final model.

## 3. Results

### 3.1. Description of the Sample

The mean age was 23.1 (+/−7.4) years. The characteristics of the sample are shown in Table 2 and Table 3. *p*-values ≤ 0.05 are in bold in Table 2 and Table 3. A description of the sample according to the type of ED is available in Appendix A.

### 3.2. Prevalence of Food Addiction

Of the 195 patients included in the study, 163 displayed “FA” (83.6%) and 32 exhibited “No FA” at inclusion. There was no significant difference (*p* = 0.067) between the prevalence of FA in the total sample according to the version of YFAS used (1.0 or 2.0). The prevalence of FA according to ED diagnosis was 61.5% for AN-R, 87.9% for AN-BP, 97.6% for BN and 93.3% for BED diagnoses. FA prevalence was significantly different across ED diagnoses (*p* < 0.001). There was no significant difference between the prevalence of FA according to YFAS version for AN-R (*p* = 0.129) and AN-BP (*p* = 0.948). For BN and BED diagnoses, the number of patients was insufficient to conduct a comparison analysis. Regarding the presence of recurrent episodes of binge eating (at least once a week), patients (n = 114, 58.5%) with this characteristic (82 with BN diagnosis, 15 with BED diagnosis and 17 with AN-BP diagnosis) were significantly more likely to have FA (*p* < 0.001).

### 3.3. Frequency of YFAS Criteria

Regarding the whole sample, the most prevalent criteria were (i) clinically significant impairment or distress in relation to food (90.8%); (ii) craving (79.2%); and (iii) persistent desire or repeated unsuccessful attempts to cut down (78.5%). Table 4 shows the percentage of each YFAS criterion met in the total sample, as well as for each type of ED. “Clinically significant impairment or distress in relation to food” was one of the most prevalent diagnostic criteria regardless of the type of ED, but each type of ED was associated with specific criteria: “Use in physically hazardous situations” with ED associated with under- or overweight, “craving” with ED characterized by recurrent episodes of binge eating.

### 3.4. Bivariate Comparison of the “FA” Group and the “No FA” Group

No differences were observed in the sociodemographic data and severity of ED, but several significant differences were observed for ED and other clinical characteristics. Table 2 and Table 3 show the results of the comparisons of the two groups.

### 3.5. Factors Associated with Food Addiction

Sample sizes for BN and BED in the “no FA” group were too small. We therefore have chosen to include the variable “recurrent episodes of binge eating” (at least once a week) instead of “type of ED” in the multiple regression model. Among the variables selected based on the bivariate analysis to be included as candidates in the multiple regression model, high correlations were found, leading to the exclusion of four variables (“EDI-feeling of ineffectiveness”, “EDI-impulse regulation”, “EDI-interpersonal distrust” and “EDI-social insecurity”) from the multivariate analysis.

Following multiple logistic regression, only three variables remained independently associated with FA: presence of recurrent episodes of binge eating (OR = 28.2), lower MROAS total score (OR = 0.67) corresponding to a higher severity of ED, and higher “EDI-interoceptive awareness” score (OR = 1.22) corresponding to a higher lack of interoceptive awareness (Table 5). The Hosmer–Lemeshow goodness-of-fit test was non-significant (*p* = 0.60; Chi-squared = 6,460 and df = 8), showing that the final model was well calibrated. The area under the ROC curve was 0.91, indicating that the model discriminated well between patients who had FA (N = 165) and those who did not have FA (N = 32).

## 4. Discussion

### 4.1. Main Results

The aim of this study was to estimate the prevalence of FA in a sample of ED patients. We also aimed to assess the most commonly met criteria of the YFAS and to determine the factors associated with the presence of FA.

First, our study confirmed the hypothesis of a strong link between an FA diagnosis according to YFAS and an ED diagnosis, with a prevalence of 83.6% in a cohort of 195 patients suffering from ED. These results are fully in line with previous work, with a prevalence ranging from 70 to 90% (1). The data seem to demonstrate an overlap between FA and ED, with a gradient according to the type of ED: prevalence of FA appeared to be important in BN (97.6%) and BED (93.3%), as other studies had previously demonstrated [21,47,48,49], as well as in AN-R (61.5%) and AN-BP (87.9%). The umbrella term ED encompasses a broad spectrum of disorders, with AN at one end and BED at the other, and includes BN and other specified feeding and eating disorders (OSFEDs). The high FA prevalence in BED, BN patients and, to a lesser extent, AN-BP patients is not surprising given that EDs defined by the presence of binge eating share behavioral, clinical and neurobiological characteristics with other types of addictive disorders [50,51,52,53,54,55,56,57,58,59,60,61]. However, conceptualizing AN-R as overlapping with FA is somewhat more debatable. For example, Barbarich-Marsteller et al. (2011) stated that AN is not an addiction [62]. Indeed, people with AN-R seem not to be addicted to food but quite the opposite, i.e., addicted to food deprivation, and they show real determination instead of losing control. In their recent paper, Mallorquí-Bagué et al. (2020) concluded that patients with AN exhibited a successful down-regulation of food craving, despite the presence of food addiction symptomatology [63]. In the present study, we found a prevalence of FA in patients with AN-R that was far more substantial than in nonclinical samples (0 to 25%) [11]. Our results are in line with those of Granero et al. [21] and Wolz et al. [20] The YFAS was built to screen addictive symptoms, but it is important to note that the “object” of addiction is not clearly specified. Despite the fact that the YFAS putatively explores eating behaviors toward specific hyperpalatable foods high in fat and/or sugar (i.e., pizza, sweets, soda, chips, etc.), it has been debated whether any of these foods comprise different “substances” [4]. Then, the substance of abuse is not defined, which raises an important question in the explanation of addictive-like eating: are the addictive properties intrinsic to some foods or associated with eating behavior? As has been shown for rodents and humans, certain types of food, such as high sugar and high fat palatable foods, have rewarding properties [4]. From an evolutionary point of view, these foods promote survival by increasing the motivation to eat nutrients with a substantial energy value. In our societies, which are characterized by easy access to highly palatable food, these specific properties could overwhelm cognitive inhibition and homeostatic mechanisms and lead to overweight [4]. However, as explained by Hebebrand et al. [4], characterizing a food or nutriment as an addictive substance implies that it has intrinsic addictive properties with the capacity to make vulnerable individuals addicted to it. In their recent article, Fletcher and Kenny wrote that no clear consensus has yet emerged on the validity of the concept of food addiction, and they presented arguments and counterarguments [64]. Regarding human research, Ahmed et al. concluded in their review that sugar and sweet rewards could not only substitute for addictive drugs such as cocaine, but also could potentially be more rewarding and attractive [65]. However, apart from caffeine, human research has found no clear evidence that any specific food, ingredient, micronutrient or combination is addictive and thus that some individuals would crave some foods akin to ingesting a specific substance. Hebebrand et al. therefore proposed the term eating addiction rather than food addiction to better capture eating addiction-related disorders [4], going beyond the substance-based view assumed in the YFAS. This term eating addiction might partially help explain why the prevalence of FA is so high in patients with AN-R because of their relationship with food. The notable prevalence of FA in AN-R but also in AN-BP patients might then be linked with natural consequences of chronic food deprivation, as shown in the Minnesota Semistarvation Experiment, which resulted in preoccupation with food and conversations centered around food, recipes and food production among healthy volunteers submitted to severe and prolonged dietary restriction [66]. However, these results are also in line with the clinical experience of AN (AN-R as AN-BP) patients showing restrictive eating behaviors to combat impulses of hunger and a loss of control over eating. The classic shift from restriction to binge eating is one argument, among others, that supports this notion. In that way, a study using functional magnetic resonance imaging (fMRI) in women recovered from AN-R showed an increased neural response to pleasant food stimuli in the ventral striatum, a brain region implicated in the motivational salience of stimuli [67]. According to the authors, these results support the idea that AN-R patients may restrict their eating in order to control exposure to food stimuli because of a hypersensitive neural response to them. However, it is difficult to determine whether this neural dysfunction is a stable trait characteristic preceding the development of AN-R, supporting the theory of addiction-like eating tendencies in AN-R patients, or a scar effect. Longitudinal studies are needed to answer that question.

Second, the analysis of each criterion revealed that the most prevalent ones in our sample were (i) “clinically significant impairment or distress in relation to food” (90.8%); *(ii) “craving”* (79.2%); and (iii) “persistent desire or repeated unsuccessful attempts to cut down” (78.5%). Previous studies have found similar results, with the criteria “clinically significant impairment or distress in relation to food” and “persistent desire or repeated unsuccessful attempts to cut down” being the most important criteria in ED patients [21,47,48]. Nevertheless, *“craving”* has not been evaluated in previous studies because this criterion was not present in the first version of the YFAS. The frequency of the first criterion is not surprising given that a significant impairment or distress in relation to food is a core feature in ED. The importance of craving is in line with the evolution in the addiction-related diagnostic criteria according to the DSM: whereas the presence of tolerance or withdrawal symptoms was necessary to confirm a diagnosis of Alcohol Dependence in the DSM-III [68], it was no longer the case with the publication of the DSM-IV [8]. In the DSM-5 [27], craving appeared as a new diagnostic criterion and has been progressively viewed as a relevant and core symptom in addiction. In our study, we observed that “classic” symptoms such as tolerance and withdrawal were not among the three most frequent criteria fulfilled in our sample, irrespective of the type of ED. This finding is in line with the conceptual evolution of the definition of addictive disorders but again calls into question the relevance of the substance-based model of FA. However, craving was the second most fulfilled criterion in AN-BP and BN patients, suggesting once again an overlap with addictive disorders, and the fourth most fulfilled criterion in AN-R patients, which might indicate a natural response to chronic food restriction but might also be linked to a natural affinity for eating as mentioned previously. Regarding the frequency of the criterion “persistent desire or repeated unsuccessful attempts to cut down”, it is in line with previous studies conducted with both clinical and general population samples, in which this criterion was the most frequently endorsed FA symptom [48]. It reflects a behavioral control failure typically observed in EDs as well as in addictive disorders. Furthermore, it is noteworthy that this criterion was the third most frequently fulfilled in AN-R patients (64.6%), possibly due to a misunderstanding related to their subjective feeling of eating too much. Overall, the relevance of modeling FA criteria based on SUD criteria to better conceptualize overeating has been debated [69], and should be considered with caution when energy intake is restricted, as in AN-R, but also in AN-BP and, to a lesser extent, in BN. It is thus difficult to conclude firmly that these criteria can be considered symptoms of FA.

Third, the presence of FA in our sample appeared to be independently correlated with three variables: illness severity, the presence of binge-eating episodes and a more pronounced lack of interoceptive awareness assessed by the EDI-2. As noted in previous studies, the presence of addiction-related symptoms is associated with a more severe eating pathology and psychopathology among ED patients [11,20,21,47]. The association between FA and the presence of binge eating episodes is also in line with previous research. In the study conducted by Granero et al. [21], higher dimensional scores in the YFAS were associated with the number of binge episodes per week (and not with the number of purging behaviors per week). More generally, several authors have highlighted that FA represents an extreme state of overeating (with a correlation between the number of YFAS symptoms and BMI in most of the studies [11]) and a more severe variant of BED [52,70,71], since binge eating has been consistently correlated with YFAS scores [7,17,71,72]. Given this, a high-risk population might be identified, and made-to-measure treatment approaches might be proposed based on the existence of FA. Indeed, a potential therapeutic implication would be to tailor SUD interventions to individuals exhibiting binge-eating episodes. This could involve motivational interviewing, psychoeducational programs, cognitive behavioral therapy to cope with cravings and cognitive remediation focusing on executive function and inhibitory control, classically proposed for SUDs. Moreover, the present findings support the development of drug therapy targeting the reward circuitry such as mu opiate receptor antagonists, for these patients [73]. Regarding the greater lack of interoceptive awareness found in the patients with FA, this could constitute a bias suggesting that FA may have been overestimated, especially in patients with AN-R. The lack of interoceptive awareness reflects one’s lack of confidence in recognizing and accurately identifying emotions and sensations of hunger or satiety and was labeled fundamental to AN by Bruch and Selvini-Palazzoli [74,75,76]. Thus, some items could have been coded as positive by patients because of difficulties in recognizing sensations of hunger or satiety. We could also consider that a lack of interoceptive awareness might truly predict FA. ED patients with FA might exhibit a different profile than ED patients without FA. Therefore, a more specific treatment program, notably based on body-oriented psychotherapy aimed at improving interoceptive skills, could be proposed according to the presence of FA. In our sample, FA was not associated with the expected factors that are typically correlated with FA. This might be due to the ED sample heterogeneity and to the existence of associations between these factors and certain types of EDs, as found in the literature [77,78,79,80,81,82,83], displaying comorbidity and personality traits shared between FA and ED.

According to the DSM-5 [27], there is an overlap between Substance-related and Addictive Disorders and Feeding and Eating Disorders, given that “control” plays a major role in these two categories of disorders. Whereas “impaired control” (which may reflect impairments in brain inhibitory mechanisms) appears to be a key feature in SUDs, a “sense of lack of control over eating during the (binge eating) episode” is presented in the DSM as more central in both BN and BED [27]. Thus, in BN and BED, as noted by Hebebrand et al. [4], the focus is made by the DSM on subjective feelings of the loss of control. The importance of FA in ED patients questions the pertinence of this distinction between objective impaired control in the field of addictive disorders on the one hand and a subjective sense of lack of control in the field of Feeding and Eating Disorders on the other hand. Some studies have suggested that disturbances in the inhibitory control pathway, occurring in particular rewarding conditions, may favor ED, in particular BED and BN [6,84,85]. Moreover, in AN patients, the literature has also suggested the pivotal role of the reward system in the context of exposure to particular stimuli, such as underweight stimuli for patients presenting acute AN [86], that support theories of starvation dependence, and food stimuli for patients presenting recovered AN as previously cited [67], that supports a particular affinity for eating, which persists even after starvation. That being said, in addictive disorders, subjective feelings about the “object” of addiction need to be taken into account as much as the objective impaired control. In that sense, a study demonstrated that the subscale of the Food Cravings Questionnaire-Trait that assesses the anticipation of positive reinforcement that may result from eating had negatively predicted FA symptoms, contrary to the other subscales [87]. According to the authors, people with FA symptoms may want craved foods but were also aware that the food will not make them feel better. Similarly, they experienced feelings of guilt after giving in to cravings. According to the authors, these results illustrate the ambivalence associated with food craving experiences, which seem to be especially important in individuals with addictive-like eating behaviors. Then, patients with FA might experience craving for food associated with a substantial sentiment of ambivalence and guilt. It is noteworthy that these clinical aspects are particularly observed in AN.

In this way, an integrative treatment approach inspired on the one hand by classic ED treatment based on nutrition rehabilitation, body-oriented psychotherapy, and cognitive therapy aimed at reducing cognitive distortions about eating, body shape and weight and on the other hand by traditional SUD treatment as cited above should be developed for ED patients with FA, taking into account the presence of addictive tendencies.

### 4.2. Strengths and Weaknesses

The results must be viewed in the context of some limitations. First, compared with the AN and BN groups, the BED group was small (*n* = 15), which could have minimized the power of the study. Second, the cognitive distortions that usually affect ED patients, notably AN patients, could have skewed the way they answered the questionnaires. Some items, such as “I continued to eat certain foods even though I was no longer hungry”, “I spend a lot of time feeling sluggish or fatigued from overeating”, “I felt so bad about overeating that I didn’t do other important things”, *or* “I didn’t do well at work or school because I was eating too much”, could have been coded as positive by the patients because of difficulties in recognizing sensations of hunger or satiety and because of particular beliefs about eating. However, as previously stated, the YFAS does not measure objective overeating but a particular relationship with food and eating; thus AN patients could satisfy the criteria for FA even if it is quite difficult to determine whether this tendency stems from starvation or a natural affinity for eating. Other limitations include the cross-sectional design of the study and the definition of the FA concept in itself, which is still a debated topic (i.e., does the YFAS truly measure what it is designed to measure?).

These limits are compensated by the strengths of the study. First, we want to emphasize the sample size. Two hundred and one patients were recruited, and such a large number allowed for a good representativeness of patients seeking treatment for ED. Moreover, ED diagnoses were established by structured clinical interviews and were based on DSM criteria. All patients were assessed at the beginning of the care in our specialized department. Finally, to the best of our knowledge, only a few studies have evaluated FA in AN [20,21], and we provided original results.

### 4.3. Perspectives

In showing an overlap between ED and FA, this study allows consideration of ED, including AN-R, from an addictive perspective, thus paving the way for therapeutic management that draws from those proposed for addictive disorders. Given that patients with ED and FA exhibit a different profile than patients with ED and no FA, tailor-made treatment might be proposed based on the existence of FA. Because the object of addiction is not clearly defined in the YFAS, the relevance of modeling its criteria on diagnostic criteria for SUD can be questioned, and further studies are needed to evaluate the intrinsic nature of some food addictive properties. The term eating addiction rather than FA might be most appropriate because it includes the behavioral component of the disorder. It would be of interest in further studies to specifically assess eating addiction in ED samples, with a specific scale such as the Addiction-like Eating Behaviour Scale [88]. This could be considered a clinical entity that needs to be better characterized.

## Figures and Tables

**Figure 1 nutrients-12-01897-f001:**
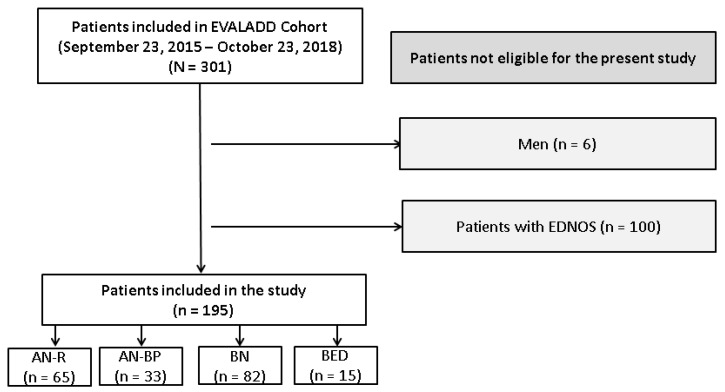
Flow chart of patient selection. AN-R: anorexia nervosa restricting type; AN-BP: anorexia nervosa binge eating/purging type; BN: bulimia nervosa; BED: binge-eating disorder; EDNOS: eating disorders not otherwise specified; *n*: number of patients included or excluded at each step of the inclusion process.

**Table 1 nutrients-12-01897-t001:** Discrepancies and similarities between food addiction (FA) and all types of eating disorders (Eds): comparison of different criteria between each of the types of disorders.

	AN-R	AN-BP	BN	BED	FA
Binge eating episodes		X	Xx	Xx	
Excessive food consumption		X	Xx	Xx	X
Sense of lack of control/loss of control of eating		X	Xx	Xx	X
Intense fear of gaining weight/self-evaluation unduly influenced by body shape and weight	Xx	Xx	Xx	Associated feature	
Restriction in food intake	Xx	Xx	Associated feature		
Restriction of energy intake/recurrent behaviors that interfere with weight gain/inappropriate compensatory behaviors	Xx (dieting, fasting, excessive exercise)	Xx (dieting, fasting, excessive exercise/purging behaviors)	Xx (excessive exercise, purging behaviors, or fasting)		
Obsessions related to food	Associated feature	Associated feature	Associated feature	Associated feature	X
Distress in relation to food	Associated feature	Associated feature	Associated feature	Associated feature	Xx
Social and/or professional consequences	Associated feature	Associated feature	Associated feature	Associated feature	X

AN-R: anorexia nervosa restricting type; AN-BP: anorexia nervosa binge eating/purging type; BN: bulimia nervosa; BED: binge-eating disorder; FA: food addiction; X: diagnostic feature; Xx: necessary diagnosis feature; Associated feature: symptom classically associated according to the Diagnostic and Statistical Manual (DSM) and/or the literature.

**Table 2 nutrients-12-01897-t002:** Description of the sample and comparison of the patients with “Food Addiction” or “No Food Addiction” (*n* = 195)—Sociodemographic and Eating Disorder characteristics.

	Entire Sample (*N* = 195)	“Food Addiction” (*n* = 163)	“No Food Addiction” (*n* = 32)	*p*-Value	Statistical Test
*n (%) or m (sd)*					
**Sociodemographic characteristics**					
***Age*** (years)	23.1 (7.4)	23.3 (7.8)	22.1 (5.2)	0.903	Wilcoxon
**Eating disorder characteristics**					
***Type of ED***				**<0.001**	Chi^2^
AN-R	65 (33.3%)	40 (24.5%)	25 (78.1%)		
AN-BP	33 (16.9%)	29 (17.8%)	4 (12.5%)		
BN	82 (42.1%)	80 (49.1%)	2 (6.3%)		
BED	15 (7.7%)	14 (8.6%)	1 (3.1%)		
***Recurrent episodes of binge eating (yes)***	114 (58.5%)	110 (67.5%)	4 (12.5%)	**<0.001**	Chi^2^
***Age of disease onset*** (years)	15.9 (5.0)	15.6 (5.1)	17.4 (4.0)	**0.006**	Wilcoxon
***Disease duration*** (years)	7.2 (7.6)	7.6 (7.9)	4.8 (5.3)	**0.050**	Wilcoxon
***Severity of ED*** (MROAS total score)	6.4 (2.0)	6.3 (2.0)	6.9 (1.9)	0.111	Student’s t
***Dimensions associated with ED*** (EDI-2)					
Ineffectiveness	13.4 (7.0)	14.0 (7.0)	10.1 (6.0)	**0.004**	Student’s t
Interoceptive awareness	13.6 (6.8)	14.8 (7.4)	6.5 (4.7)	**<0.001**	Wilcoxon
Asceticism	8.5 (4.5)	8.9 (4.5)	6.6 (3.5)	**0.007**	Student’s t
Drive for thinness	15.3 (5.1)	16.0 (4.4)	11.6 (6.5)	**<0.001**	Wilcoxon
Bulimia	8.2 (6.7)	9.4 (6.4)	2.0 (3.4)	**<0.001**	Wilcoxon
Body dissatisfaction	18.0 (7.2)	18.9 (7.0)	13.3 (6.4)	**<0.001**	Student’s t
Perfectionism	7.3 (4.5)	7.7 (4.4)	5.0 (3.7)	**<0.001**	Student’s t
Interpersonal distrust	7.7 (4.5)	7.9 (4.7)	6.7 (3.6)	0.165	Student’s t
Maturity fears	7.9 (5.9)	8.1 (5.9)	6.8 (6.0)	0.263	Student’s t
Impulse regulation	8.4 (6.6)	9.2 (6.7)	4.4 (3.8)	**<0.001**	Wilcoxon
Social insecurity	9.9 (4.8)	10.3 (4.8)	7.6 (4.2)	**0.003**	Student’s t

**%**: percentage; m: mean; sd: standard deviation; AN-BP: anorexia nervosa binge-eating/purging type; AN-R: anorexia nervosa restricting type; BED: binge eating disorder; BN: bulimia nervosa; ED: eating disorder; EDI: Eating Disorders Inventory; MROAS: Morgan–Russell Outcome Assessment Schedule; sd: standard deviation; YFAS: Yale Food Addiction Scale.

**Table 3 nutrients-12-01897-t003:** Description of the sample and comparison of the patients with “Food Addiction” or “No Food Addiction” (*n* = 195)—Other clinical characteristics.

	Entire Sample (*N* = 195)	“Food Addiction” (*n* = 163)	“No Food Addiction” (*n* = 32)	*p*-Value	Statistical Test
		*n (%) or m (sd)*			
**Comorbidities (current or past)**					
Mood disorders (MINI)	156 (80.0%)	135 (83.8%)	21 (65.6%)	**0.026**	Chi^2^
Anxiety disorders (MINI)	141 (72.3%)	124 (76.1%)	17 (53.1%)	**0.008**	Chi^2^
Psychotic syndrome (MINI)	12 (6.2%)	10 (6.0%)	2 (5.9%)	0.981	Chi^2^
Addictive disorders (MINI and MIDI)	90 (46.2%)	79 (48.5%)	11 (34.4%)	0.144	Chi^2^
ADHD in childhood (WURS-C)	66 (33.8%)	63 (38.7%)	3 (9.4%)	**0.001**	Chi^2^
**Impulsivity**					
UPPS-Urgency	10.5 (3.0)	10.8 (2.9)	9.3 (3.1)	**0.009**	Student’s t
UPPS-Premeditation (lack)	7.5 (2.5)	7.6 (2.6)	7.2 (2.0)	0.438	Student’s t
UPPS-Perseverance (lack)	7.4 (2.9)	7.6 (2.9)	6.8 (2.6)	0.140	Student’s t
UPPS-Sensation seeking	9.8 (3.2)	9.8 (3.1)	10.0 (3.5)	0.803	Student’s t
**Temperament Comorbidities (current or past)**					
TCI-Novelty seeking	40.8 (19.3)	41.9 (19.0)	35.3 (19.7)	0.077	Student’s t
TCI-Harm avoidance	74.2 (20.6)	75.1 (20.1)	69.4 (22.7)	0.150	Student’s t
TCI-Reward dependence	60.4 (17.5)	59.4 (17.9)	65.1 (15.1)	0.094	Student’s t
TCI-Persistence	72.6 (28.7)	72.0 (28.8)	75.6 (28.2)	0.518	Student’s t
**Attachment**					
RSQ-Secure	2.7 (0.6)	2.7 (0.6)	2.7 (0.5)	0.896	Student’s t
RSQ-Fearful	2.9 (0.6)	2.9 (0.6)	2.7 (0.6)	0.091	Student’s t
RSQ-Preoccupied	2.6 (0.7)	2.6 (0.6)	2.4 (0.7)	0.106	Student’s t
RSQ-Dismissing	3.3 (0.8)	3.3 (0.8)	3.1 (0.9)	0.172	Student’s t
**Life events**					
History of physical abuse	20 (10.3%)	20 (12.3%)	0	-	-
History of sexual abuse	27 (13.8%)	24 (14.7%)	3 (9.4)	0.423	Chi^2^

%: percentage; m: mean; sd: standard deviation; ADHD: attention-deficit/hyperactivity disorder; MIDI: Minnesota Impulsive Disorders Interview; MINI: Mini International Neuropsychiatric Interview; RSQ: Relationship Scales Questionnaire; sd: standard deviation; TCI: Temperament and Character Inventory; UPPS: Impulsive behavior scale; WURS-C: Wender Utah Rating Scale-Child.

**Table 4 nutrients-12-01897-t004:** Percentage of each YFAS criterion met for ED patients and according to the type of ED.

	Total Sample (*N* = 195)		AN-R (*n* = 65)		AN-BP (*n* = 33)		BN (*n* = 82)		BED (*n* = 15)	
YFAS Criteria	Valid *N*	Number of patients (%) with positive YFAS criteria	Valid *N*	Number of patients (%) with positive YFAS criteria	Valid *N*	Number of patients (%) with positive YFAS criteria	Valid *N*	Number of patients (%) with positive YFAS criteria	Valid *N*	Number of patients (%) with positive YFAS criteria
1- Loss of control	195	127 (65.1%)	65	16 (24.6%)	33	21 (63.6%)	82	76 (92.7%)	15	14 (93.3%)
2- Persistent desire or repeated unsuccessful attempts to cut down	195	153 (78.5%)	65	42 (64.6%)	33	22 (66.7%)	82	75 (91.5%)	15	14 (93.3%)
3- Much time spent	195	110 (56.4%)	65	18 (27.7%)	33	17 (51.5%)	82	61 (74.4%)	15	14 (93.3%)
4- Craving *	77	61 (79.2%)	23	14 (60.9%)	16	12 (75.0%)	34	33 (97.1%)	4	2 (50.0%)
5- Continued used despite social or interpersonal problem *	77	54 (70.1%)	23	10 (43.5%)	16	10 (62.5%)	34	30 (88.2%)	4	4 (100%)
6- Impaired daily functioning *	77	46 (59.7%)	23	8 (34.8%)	16	8 (50.0%)	34	28 (82.4%)	4	2 (50.0%)
7- Important activities given up	195	141 (72.3%)	65	36 (55.4%)	33	23 (69.7%)	82	69 (84.2%)	15	13 (86.7%)
8- Use in physically hazardous situations *	77	56 (72.7%)	23	17 (73.9%)	16	12 (75.0%)	34	23 (67.7%)	4	4 (100%)
9- Use despite knowledge of adverse consequences	195	100 (51.3%)	65	25 (38.5%)	33	16 (48.5%)	82	49 (59.8%)	15	10 (66.7%)
10- Tolerance	195	95 (48.7%)	65	18 (27.7%)	33	17 (51.5%)	82	52 (63.4%)	15	8 (53.3%)
11- Withdrawal symptoms	195	119 (61.0%)	65	19 (29.2%)	33	23 (69.7%)	82	66 (80.5%)	15	11 (73.3%)
12- Clinically significant impairment or distress	195	177 (90.8%)	65	52 (80.0%)	33	31 (93.9%)	82	80 (97.6%)	15	14 (93.3%)

* Criteria that are present only in the second version of the YFAS (modeled on DSM-5 criteria), assessed in only 77 ED patients, 23 AN-R patients, 16 AN-BP patients, 33 BN patients, and 4 BED patients. %: percentage; AN-BP: anorexia nervosa binge-eating/purging type; AN-R: anorexia nervosa restricting type; BED: binge-eating disorder; BN: bulimia nervosa; ED: eating disorder; YFAS: Yale Food Addiction Scale.

**Table 5 nutrients-12-01897-t005:** Multiple logistic regression analysis (final model)—factors associated with “food addiction” (*N* = 195).

Variables	OR	CI_95%_ (OR)	*p*-Value
MROAS total score	0.67	[0.50; 0.89]	<0.01
EDI-2 interoceptive awareness	1.22	[1.10; 1.34]	<0.001
Recurrent episodes of binge eating (yes)	28.20	[7.00; 113.7]	<0.0001

EDI: Eating Disorder Inventory; CI_95%_: 95% confidence interval; MROAS: Morgan–Russel Outcome Assessment Schedule; OR: odds ratio.

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
