# Peer review of "Food Addiction among Female Patients Seeking Treatment for an Eating Disorder: Prevalence and Associated Factors"

_nutrients, 2020, doi:10.3390/nu12061897_

Round 1

Reviewer 1 Report

The study reports a high prevalence of YFAS-diagnosed ‘food addiction’ within patients with bulimia, restrictive type and binge-purge type anorexia, and binge eating disorder. YFAS-diagnosed food addiction was also independently associated with recurrent binge eating episodes, eating disorder severity, and low interoceptive awareness.

Overall, the paper is well written and provides novel insight into the prevalence of YFAS food addiction and disordered eating. However, the conclusion that eating disorders can be viewed as addictive disorders assumes that the YFAS provides a valid assessment of food addiction. This assumption is problematic because the validity of the YFAS depends upon the extent to which a) certain types of overeating represent a substance-based addiction, and b) the SUD criteria can be appropriately applied to the assessment of food addiction. As such, the validity of the YFAS is widely debated and so the authors should take this into consideration when interpreting their findings. In particular, the authors should consider the following:

1)the high prevalence of YFAS-diagnosed food addiction within people with restrictive type anorexia is unlikely to reflect objective overeating but rather an unhealthy relationship with food. The authors have recognised this in their Discussion, but overlook an alternative interpretation; specifically, that the YFAS identifies people with an unhealthy relationship with food, but who do not necessarily have addiction-like eating tendencies. Further to this point, ‘cravings’ and isolated bouts of overeating reported by people with AN-R is likely to be a natural (and healthy) response to chronic food restriction, rather than addiction-like processes.

2) In this study, the three most prevalent YFAS criteria amongst participants with eating disorders were 'clinical impairment/distress’, ‘cravings’, and ‘persistent desire’, but to what extent can these characteristics be considered symptoms of ‘food addiction’? As mentioned above, food cravings and desire are likely a natural response to attempts at food restriction, and fulfilment of the ‘clinically significant impairment’ criterion is not surprising given that this was a clinical population. The problems associated with applying the SUD criteria to overeating has been discussed previously (e.g. Ziauddeen Farooqi, & Fletcher, 2012) and I recommend that the authors consider these limitations when interpreting these findings. Furthermore, the fact that ‘withdrawal’ and ‘tolerance’ were not commonly reported provides evidence against the substance-based model of food addiction and, in doing so, questions the construct validity of the YFAS.

3) A further issue that should be clarified is whether fulfilment of the YFAS criterion for food addiction provides a clinically meaningful diagnosis. If the vast majority of people with eating disorders fulfil this criterion, then how is ‘food addiction’ a distinct concept that should be treated differently from other eating disorders? The authors find evidence that those with YFAS diagnosis have a more severe disorder, but the extent to which YFAS-diagnosed food addiction is qualitatively different from other types of eating disorders remains unclear. This is important because, if food addiction is to be considered within future editions of the DSM, then it would be necessary to show how it is distinct from other eating disorders, both with regards to behavioural and cognitive manifestations and treatment approaches.

4) The authors suggest the need to reclassify certain types of overeating as an ‘eating addiction’, rather than ‘food addiction’. However, it should be recognised that the YFAS does not assess ‘eating addition’ but rather assumes a substance-based view of food addiction (which, as the authors point out, is not well supported by the literature). Alternative measures have been developed for assessing eating addiction (e.g. the Addiction-like Eating Behaviour Scale, Ruddock et al., 2017) and these might provide more valid assessment of addictive patterns of eating.

To summarise, I recommend that the findings should be interpreted in light of the limitations of the YFAS and substance-based account of food addiction. The authors should avoid making the assumption that fulfilment of the YFAS criteria necessarily provides evidence for ‘food addiction’, and should discuss the extent to which YFAS-diagnosed food addiction represents a distinct clinical disorder.

Other (more minor) issues include:

- It is unclear what is meant by the 'qualitative variables' described in section 2.5.

- In Tables 1 and 2, please provide clarification as to which values represent means, and which represent frequencies.

Author Response

The study reports a high prevalence of YFAS-diagnosed ‘food addiction’ within patients with bulimia, restrictive type and binge-purge type anorexia, and binge eating disorder. YFAS-diagnosed food addiction was also independently associated with recurrent binge eating episodes, eating disorder severity, and low interoceptive awareness.

Overall, the paper is well written and provides novel insight into the prevalence of YFAS food addiction and disordered eating.

Answer: We thank the reviewer for his/her compliments.

However, the conclusion that eating disorders can be viewed as addictive disorders assumes that the YFAS provides a valid assessment of food addiction. This assumption is problematic because the validity of the YFAS depends upon the extent to which a) certain types of overeating represent a substance-based addiction, and b) the SUD criteria can be appropriately applied to the assessment of food addiction. As such, the validity of the YFAS is widely debated and so the authors should take this into consideration when interpreting their findings.

Answer: We agree with the reviewer, and we made some modifications as presented below to improve clarity.

In particular, the authors should consider the following:

1) the high prevalence of YFAS-diagnosed food addiction within people with restrictive type anorexia is unlikely to reflect objective overeating but rather an unhealthy relationship with food. The authors have recognised this in their Discussion, but overlook an alternative interpretation; specifically, that the YFAS identifies people with an unhealthy relationship with food, but who do not necessarily have addiction-like eating tendencies. Further to this point, ‘cravings’ and isolated bouts of overeating reported by people with AN-R is likely to be a natural (and healthy) response to chronic food restriction, rather than addiction-like processes.

Answer: We thank the reviewer for mentioning the natural dimension of food craving in AN. Following this comment, we have revised paragraphs and sentences that describe this aspect and also discuss the relationship with food and the presence of addiction-like eating tendencies in AN patients (lines 453 to 468; lines 488 to 489; lines 580 to 581 in tracked changes version of the manuscript / lines 420 to 435, lines 455 to 456, lines 542 to 543 in clean version of the manuscript).

2) In this study, the three most prevalent YFAS criteria amongst participants with eating disorders were 'clinical impairment/distress’, ‘cravings’, and ‘persistent desire’, but to what extent can these characteristics be considered symptoms of ‘food addiction’? As mentioned above, food cravings and desire are likely a natural response to attempts at food restriction, and fulfilment of the ‘clinically significant impairment’ criterion is not surprising given that this was a clinical population. The problems associated with applying the SUD criteria to overeating has been discussed previously (e.g. Ziauddeen Farooqi, & Fletcher, 2012) and I recommend that the authors consider these limitations when interpreting these findings.

Answer: We thank the reviewer for mentioning the article. Following the comment, we added a sentence describing these limitations (lines 497 to 500 in tracked changes version of the manuscript/ lines 462 to 465 in clean version of the manuscript) and the reference to this article (line 498 in tracked changes version of the manuscript/ line 463 in clean version of the manuscript).

Furthermore, the fact that ‘withdrawal’ and ‘tolerance’ were not commonly reported provides evidence against the substance-based model of food addiction and, in doing so, questions the construct validity of the YFAS.

Answer: We thank the reviewer for this interesting comment. We have discussed this point (lines 484 to 486 in tracked changes version of the manuscript/ lines 451 to 452 in clean version of the manuscript).

3) A further issue that should be clarified is whether fulfilment of the YFAS criterion for food addiction provides a clinically meaningful diagnosis. If the vast majority of people with eating disorders fulfil this criterion, then how is ‘food addiction’ a distinct concept that should be treated differently from other eating disorders? The authors find evidence that those with YFAS diagnosis have a more severe disorder, but the extent to which YFAS-diagnosed food addiction is qualitatively different from other types of eating disorders remains unclear. This is important because, if food addiction is to be considered within future editions of the DSM, then it would be necessary to show how it is distinct from other eating disorders, both with regards to behavioural and cognitive manifestations and treatment approaches.

Answer: On the basis of our results, we are not able to go beyond in the discussion, concerning issues of nosography (specificity of the FA entity). Following the comment, we added or expanded phrases concerning treatment aspects (lines 513 to 519; line 527; lines 562 to 566 in tracked changes version of the manuscript/ lines 478 to 483; line 492; lines 525 to 529 in the clean version of the manuscript).

4) The authors suggest the need to reclassify certain types of overeating as an ‘eating addiction’, rather than ‘food addiction’. However, it should be recognised that the YFAS does not assess ‘eating addition’ but rather assumes a substance-based view of food addiction (which, as the authors point out, is not well supported by the literature). Alternative measures have been developed for assessing eating addiction (e.g. the Addiction-like Eating Behaviour Scale, Ruddock et al., 2017) and these might provide more valid assessment of addictive patterns of eating.

Answer: We thank the reviewer for mentioning this article. We revised a sentence (line 442 in tracked changes version of the manuscript) and added a phrase in the perspectives paragraph (lines 601-602 in tracked changes version of the manuscript/ lines 562-563 in clean version of the manuscript) with this reference (line 602 in tracked version of the manuscript/ line 563 in clean version of the manuscript).

To summarise, I recommend that the findings should be interpreted in light of the limitations of the YFAS and substance-based account of food addiction. The authors should avoid making the assumption that fulfilment of the YFAS criteria necessarily provides evidence for ‘food addiction’, and should discuss the extent to which YFAS-diagnosed food addiction represents a distinct clinical disorder.

Answer: We thank the author for this advice, and we hope that he/she will be satisfied by the modifications and clarifications made above.

Other (more minor) issues include:

- It is unclear what is meant by the 'qualitative variables' described in section 2.5.

Answer: Following the comment, we replaced “qualitative” with “categorical” (lines 290 and 297 in tracked changes version of the manuscript/ lines 266 and 273 in clean version of the manuscript) and “quantitative” with “continuous” (line 298 in tracked changes version of the manuscript/ line 273 in clean version of the manuscript) to clarify the terms.

- In Tables 1 and 2, please provide clarification as to which values represent means, and which represent frequencies.

Answer: We thank the reviewer for seeking this clarification. Following this comment, we added the percentage to represent frequencies (table 2-3).

Reviewer 2 Report

This manuscript presents the interesting issue related to the food addiction (FA) among eating disorder (ED) patients. The aim of the study were : (1) to assess the prevalence of FA among ED patients in general and according to the type of ED, (2) to investigate the most commonly fulfilled criteria of the YFAS as well as to (3) to evaluate the clinical and psychopathological correlates of FA in ED patients.

Introduction

The topic of the present paper focuses on food addiction among ED patients. Therefore, in the introduction session Authors should pay much more attention to the main topic and resign from the paragraphs related to: (a) animal models (it is not related to the objective of the study) and (b) FA and obesity (it not associated with the aim of the study as well) (lines 45-66).

In my opinion comparison between FA and ED (for showing the similarities and discrepancies) will definitely increase the value of the introduction.

Perhaps Authors may wish to include some of interesting papers on FA in their introduction:

Gordon, E.L., Ariel-Donges, A.H., Bauman, V., Merlo, L.J. (2018). What is the evidence for “food addiction?” A systematic review. Nutrients, 10(4), 477. https://doi.org/10.3390/nu10040477.

Brewerton, T.D. (2017). Food addiction as a proxy for eating disorder and obesity severity, trauma history, PTSD symptoms, and comorbidity. Eating and Weight Disorders, 22(2), 241-247. https://doi.org/ 10.1007/s40519-016-0355-8

In the introduction session there is lack of information about the (potential clinical and psychopathological) factors associated with FA. In addition, it needs to be improved to concentrate on hypotheses as well (the hypotheses should be explained).

Methods

Procedure and ethics

Page 3, line 114: Please explain what does the abbreviation EVALADD mean.

Page 3, lines 119-120: Please add more information about the semi-structured interview (EDQ? EDA? IDED-IV? MROAS?) and completed self-report questionnaires (please indicate some examples).

Page 3, line 123: Please complete the number of the approval from the local ethics committee.

Page 3, lines 132-133: Please also add the percentage of the ED patients (Sixty-seven patients (….%) were diagnosed with AN-R, 34 (…%) with AN-BP, 85 (…%) with BN and 15 (…%) with BED)

Sociodemographic characteristics

Due to the objectives of the present study (assessment of the prevalence of FA and associated factors of FA among ED patients in general and according to the type of ED), the sociodemographic data included age and gender should be presented for total ED patients and separately for each type of ED.

There is lack of explanation why among “other clinical characteristics” the following have been proposed: psychiatric comorbidities, impulsivity, temperament, attachment and history of traumatic events. The use of the methods should result from the introduction session. Therefore, I would strongly recommend Authors to sum up some fundamental (recent) data focusing on these variables and FA.

Statistical analysis

Page 6, line 323: From my point of view, descriptive statistical analysis should be conducted for the entire sample as well as for each type of ED (please see the objectives of the study).

Page 6, line 246: Please explain why p < 0.20. In statistics, a p-value less than 0.05 (typically ≤ 0.05) is statistically significant. In addition, I do not understand why “nonsignificant variables were removed one at a time at a p < 0.05”. In statistics, if the p-value is HIGHER than 0.05 (> 0.05) is not statistically significant .

The disproportion between female and male patients is VERY LARGE (97% vs 3%), therefore, from the statistical point of view, male patients should be removed from the database and their results should not be presented in this study. In this case, the title of the manuscript should be: “Food addiction among female patients seeking treatment for an eating disorder: prevalence and associated factors”.

Page 9, line 309: Please indicate Hosmer and Lemeshow goodness of fit (GOF) test (X-squared, df)

Discussion

Main results

An in-depth discussion about the VERY HIGH prevalence of FA in ED patients is needed. In my opinion, it was not sufficiently explained. Please take into consideration that the prevalence of FA was 61.7% for AN-R a s well as 85.3% for AN-BP. It is necessary to explain these results.

In my opinion, the information about animal model (lines 353-358) should be omitted (it is not related to the results of the present study).

Authors should do more to highlight their findings related to clinical and psychopathological correlates of FA in ED patients. What are the practical implications of these findings?

Some part of the discussion (e.g. about “eating addiction”, line 323) should be presented in the introduction section.

I would suggest go deeper in a discussion about three factors associated with the presence of FA (illness severity, lack of interoceptive awareness and the presence of binge-eating episodes). Please take into consideration the clinical (and therapeutic) aspects of these results.

Perspectives

In my opinion, “an overlap between ED and FA” was not sufficiently presented in the discussion part, therefore I would suggest to improve this part of the paper for better understanding of FA prevalence and association with ED.

Author Response

This manuscript presents the interesting issue related to the food addiction (FA) among eating disorder (ED) patients. The aim of the study were : (1) to assess the prevalence of FA among ED patients in general and according to the type of ED, (2) to investigate the most commonly fulfilled criteria of the YFAS as well as to (3) to evaluate the clinical and psychopathological correlates of FA in ED patients.

Answer: We thank the reviewer for his/her positive comment.

Introduction

The topic of the present paper focuses on food addiction among ED patients. Therefore, in the introduction session Authors should pay much more attention to the main topic and resign from the paragraphs related to: (a) animal models (it is not related to the objective of the study) and (b) FA and obesity (it not associated with the aim of the study as well) (lines 45-66).

Answer: We agree with the reviewer, so we made modifications as indicated below in order to focus more on the main topic. However, we think that paragraphs related to (a) the animal model and (b) FA and obesity are important for better capturing issues related to the FA concept, mentioned in the discussion. Following this comment, we shortened the paragraphs (lines 62 to 80 in tracked changes version of the manuscript/ lines 55 to 62 in clean version of the manuscript).

In my opinion comparison between FA and ED (for showing the similarities and discrepancies) will definitely increase the value of the introduction.

Answer: Following this comment, we expanded a sentence (lines 117-118 in tracked changes version of the manuscript/ lines 97-98 in clean version of the manuscript) and added a table that describes the similarities and discrepancies between FA and each type of ED (table 1).

Perhaps Authors may wish to include some of interesting papers on FA in their introduction:

Gordon, E.L., Ariel-Donges, A.H., Bauman, V., Merlo, L.J. (2018). What is the evidence for “food addiction?” A systematic review. Nutrients, 10(4), 477. https://doi.org/10.3390/nu10040477.

Brewerton, T.D. (2017). Food addiction as a proxy for eating disorder and obesity severity, trauma history, PTSD symptoms, and comorbidity. Eating and Weight Disorders, 22(2), 241-247. https://doi.org/ 10.1007/s40519-016-0355-8

Answer: We thank the reviewer for mentioning these two interesting papers. We added these references in the introduction (reference 3 and 14).

In the introduction session there is lack of information about the (potential clinical and psychopathological) factors associated with FA. In addition, it needs to be improved to concentrate on hypotheses as well (the hypotheses should be explained).

Answer: Following this comment, we completed the introduction in order to provide more information about factors associated with FA (lines 105-111 in tracked changes version of the manuscript/ lines 87-92 in clean version of the manuscript) and about our study’s hypotheses (lines 128-135 in tracked version of the manuscript/ lines 114-121 in clean version of the manuscript).

Methods

Procedure and ethics

Page 3, line 114: Please explain what does the abbreviation EVALADD mean.

Answer: As recommended, the abbreviation EVALADD was defined at its first appearance (line 159 in tracked changes version of the manuscript/ line 138 in clean version of the manuscript).

Page 3, lines 119-120: Please add more information about the semi-structured interview (EDQ? EDA? IDED-IV? MROAS?) and completed self-report questionnaires (please indicate some examples).

Answer: As mentioned in section 2.3, semi-structured interviews included the MINI and the MROAS; self-report questionnaires included the EDI-2, YFAS, UPPS, TCI-125, RS-Q, and EVE. To avoid overloading the manuscript, tools have not been added to the 2.1 paragraph, but a reference to the 2.3 paragraph has been added (lines 165-166 in tracked changes version of the manuscript/ lines 144-145 in clean version of the manuscript).

Page 3, line 123: Please complete the number of the approval from the local ethics committee.

Answer: The approval number from the local ethics committee has been added (line 170 in tracked changes version of the manuscript/ line 149 in clean version of the manuscript).

Page 3, lines 132-133: Please also add the percentage of the ED patients (Sixty-seven patients (….%) were diagnosed with AN-R, 34 (…%) with AN-BP, 85 (…%) with BN and 15 (…%) with BED)

Answer: Following this comment, we added the percentages (lines 180 to 182 in tracked changes version of the manuscript/lines 159 to 161 in clean version of the manuscript).

Sociodemographic characteristics

Due to the objectives of the present study (assessment of the prevalence of FA and associated factors of FA among ED patients in general and according to the type of ED), the sociodemographic data included age and gender should be presented for total ED patients and separately for each type of ED.

Answer: As mentioned at the end of the introduction and the beginning of the discussion, the objectives of the study were

1) To estimate the prevalence of FA among ED patients in general and according to the type of ED

2) To assess the most commonly fulfilled criteria of the YFAS among ED patients

3) To determine the clinical and psychopathological correlates of FA in ED patients.

The first two objectives have been examined both in ED patients in general and by ED type. For our last objective, it was not our intention to determine the clinical correlates of FA in ED patients according to the type of ED. Indeed, from a statistical point of view, the proportions of patients without FA according to the type of ED were too small to conduct such analysis (25 with AN-R, 4 with AN-BP, 2 with BN and 1 with BED). To be clearer, we specified the population of analysis for each objective at the end of the introduction (lines 123-126 in tracked changes version of the manuscript/ lines 109-112 in clean version of the manuscript). A table with descriptive analysis according to the type of ED is also now available as supplementary material (Table S1).

There is lack of explanation why among “other clinical characteristics” the following have been proposed: psychiatric comorbidities, impulsivity, temperament, attachment and history of traumatic events. The use of the methods should result from the introduction session. Therefore, I would strongly recommend Authors to sum up some fundamental (recent) data focusing on these variables and FA.

Answer: Following this comment, we added, as mentioned above, more information about factors associated with FA (lines 105 to 111 in tracked changes version of the manuscript/ lines 87-92 in clean version of the manuscript) and revised phrase in the introduction to briefly present the method (lines 126 to 128 in tracked changes version of the manuscript/ lines 112-113 in the clean version of the manuscript).

Statistical analysis

Page 6, line 323: From my point of view, descriptive statistical analysis should be conducted for the entire sample as well as for each type of ED (please see the objectives of the study).

Answer: As mentioned above, the objective of our study was not to determine the factors associated with the presence of FA according to the type of ED. This statistical analysis would be very interesting, but a larger patient sample would be needed. Nevertheless, we added a table with descriptive analysis according to the type of ED as supplementary material (Table S1).

Page 6, line 246: Please explain why p < 0.20. In statistics, a p-value less than 0.05 (typically ≤ 0.05) is statistically significant. In addition, I do not understand why “nonsignificant variables were removed one at a time at a p < 0.05”. In statistics, if the p-value is HIGHER than 0.05 (> 0.05) is not statistically significant.

Answer: We revised section 2.5 for clarity. For the bivariate analysis, only variables with p ≤ 0.05 are statistically significant. For these variables, p-values are in bold in tables 2 and 3.We added references for a better explanation of the multiple logistic regression (lines 305 and 307 in tracked changes version of the manuscript/ lines 280 and 282 in clean version of the manuscript). Indeed, this kind of analysis usually requires the selection of all variables with a conventional level of p< 0.20 in the bivariate analysis to take into account the influence of each variable relative to each other (Mickey, R. and Greenland, S. The impact of confounder selection criteria on effect estimation. Am J Epidemiol 1989; 129:125-37).Then, an iterative selection procedure allows for retaining only significant variables with a p-value ≤ 0.05. The stepwise technique allows us to dramatically decrease the total number of models under consideration and to produce the final model (Shtatland, E.S.; Cain, E. and Barton, M.B. The peril of stepwise logistic regression and how to escape them using information criteria and the outpout delivery system. Harvard Pilgrim Health Care, Harvard Medical School, Boston, MA).

The disproportion between female and male patients is VERY LARGE (97% vs 3%), therefore, from the statistical point of view, male patients should be removed from the database and their results should not be presented in this study. In this case, the title of the manuscript should be: “Food addiction among female patients seeking treatment for an eating disorder: prevalence and associated factors”.

Answer: Following this comment, we removed all male patients from the database (n=6), and we updated all analyses. The title has also been revised.

Page 9, line 309: Please indicate Hosmer and Lemeshow goodness of fit (GOF) test (X-squared, df)

Answer: As recommended, parameters of the Hosmer and Lemeshow goodness of fit (GOF) test (X-squared, df) have been added (line 387 in tracked changes version of the manuscript/ line 360 in clean version of the manuscript).

Discussion

Main results

An in-depth discussion about the VERY HIGH prevalence of FA in ED patients is needed. In my opinion, it was not sufficiently explained. Please take into consideration that the prevalence of FA was 61.7% for AN-R a s well as 85.3% for AN-BP. It is necessary to explain these results.

Answer: Following the comment, we developed the discussion regarding the high FA prevalence in ED patients exhibiting binge-eating episodes: “The high FA prevalence in BED, BN patients and, to a lesser extent, AN-BP patients is not surprising given that EDs defined by the presence of binge eating share behavioral, clinical and neurobiological characteristics with other types of addictive disorders [41–52]”. (lines 413 to 415 in tracked changes version of the manuscript/lines 385-388 in clean version of the manuscript).

We also revised a paragraph to discuss the high prevalence in AN patients (lines 453 to 468 in tracked changes version of the manuscript/lines 420 to 435 in clean version of the manuscript).

In my opinion, the information about animal model (lines 353-358) should be omitted (it is not related to the results of the present study).

Answer: Following this comment, we deleted the part about the animal model (lines 440 to 442 in tracked changes version of the manuscript).

Authors should do more to highlight their findings related to clinical and psychopathological correlates of FA in ED patients. What are the practical implications of these findings?

Answer: Following this comment, we developed the discussion regarding the clinical and therapeutic implications of the findings (lines 513 to 519; lines 526 to 527; lines 528 to 532; lines 562 to 566 in tracked changes version of the manuscript/ lines 478-483; lines 491-492 ;lines 493 to 497; lines 525-529 in clean version of the manuscript).

Some part of the discussion (e.g. about “eating addiction”, line 323) should be presented in the introduction section.

Answer: Following this comment, we added a sentence to present the debate about food and eating addiction in the introduction section (lines 118 to 120 in tracked changes version of the manuscript/ lines 104-105in clean version of the manuscript).

I would suggest go deeper in a discussion about three factors associated with the presence of FA (illness severity, lack of interoceptive awareness and the presence of binge-eating episodes). Please take into consideration the clinical (and therapeutic) aspects of these results.

Answer: As mentioned above, we added more information about the practical implications of the findings (lines 513 to 519; lines 526 to 527; lines 528 to 532; lines 562 to 566 in tracked changes version of the manuscript/ lines 478-483; lines 491-492 ;lines 493 to 497; lines 525-529 in clean version of the manuscript).

Perspectives

In my opinion, “an overlap between ED and FA” was not sufficiently presented in the discussion part, therefore I would suggest to improve this part of the paper for better understanding of FA prevalence and association with ED.

Answer: We made changes as mentioned above, and we hope that the reviewer will be satisfied by these modifications.

Reviewer 3 Report

Fauconnier et al discovered several clinical findings between food addiction and eating disorder, especially anorexia nervosa. Anorexia nervosa traditionally consider a disease opposite to food addiction. However, accumulating evidence shows that anorexia nervosa could share a similar mechanism with food addiction. Even though the linkage is still not clear. More clinical evidence will help to find out the association between the two diseases and prompt the researchers to conduct animal approaches to investigate the underlying mechanism. This manuscript is well written. The introduction and discussion were written well. The results section and the association analysis could be improved to complement this work. 

The description for each table could be elaborated. There are wealthy information in those tables but did not reflect in the result section.

In section 3.3, the result only covered the total sample of patients with positive YFAS criteria but left the other four categories unmentioned. Was there any specific type of patient correlate with the certain YFAS criteria? 

Table 3: Why does the validate N differ among the groups? While some of the YFAS criteria include N = 201, some only have 81? What does the bold font mean for the top row of Table 3?

Several studies discussed the anorexia nervosa and food addiction but not discussed and cited in this manuscript, including PMID 32244331; 28434177; 27502054; 21999694

Author Response

Fauconnier et al discovered several clinical findings between food addiction and eating disorder, especially anorexia nervosa. Anorexia nervosa traditionally consider a disease opposite to food addiction. However, accumulating evidence shows that anorexia nervosa could share a similar mechanism with food addiction. Even though the linkage is still not clear. More clinical evidence will help to find out the association between the two diseases and prompt the researchers to conduct animal approaches to investigate the underlying mechanism. This manuscript is well written. The introduction and discussion were written well. The results section and the association analysis could be improved to complement this work. 

Answer: First, we would like to thank you for the interest that you have expressed in our article and for the valuable comments that you were kind enough to send provide.

We have carefully reviewed and adjusted the paper, and we have included all your comments, especially in the results section.

The description for each table could be elaborated. There are wealthy information in those tables but did not reflect in the result section.

Answer: We agree with the reviewer, the manuscript present a lot of results. We tried to find a good balance between the many items of information we think it is very important to present and the ease of reading the paper. So we decided to focus on the main results in the “Results” section and to be exhaustive thanks to the tables.

In section 3.3, the result only covered the total sample of patients with positive YFAS criteria but left the other four categories unmentioned. Was there any specific type of patient correlate with the certain YFAS criteria? 

Answer: Following the comment, we have added information regarding the most prevalent diagnostic criteria according to the type of ED in the “Results” section. The small sample size of some form of ED type did not allow to perform correlation analyses (lines 358 to 362 in tracked changes version of the manuscript/ lines 329-333 in clean version of the manuscript).

Table 3: Why does the validate N differ among the groups? While some of the YFAS criteria include N = 201, some only have 81? What does the bold font mean for the top row of Table 3?

Answer: The reviewer raised a crucial. Indeed, there is a difference, due to the change of the YFAS version during the study.

As mentioned in the “Method” section, “the DSM-IV criteria were used for the initial version of the YFAS. When the fifth edition of the DSM was published, a new version of the YFAS was developed. For the present study, the French version of the initial version of YFAS was used until October 31, 2017, then replaced by the French version of YFAS 2.0 once it was validated. According to Meule and Gearhardt (2019), prevalence rates and correlates of YFAS 2.0 diagnoses are largely similar to those observed with the original YFAS.”

In consequence, participants included until October 2017 were assessed by the mean of 8 criteria and participants included after this date were assessed using 12 criteria. Each participant was assessed at least by the 8 common criteria.

This point is explained in the legend of the table.

The bold font for the top row of Table 3 was due to formatting issue, we corrected it.

Several studies discussed the anorexia nervosa and food addiction but not discussed and cited in this manuscript, including PMID 32244331; 28434177; 27502054; 21999694

Answer: We thank the reviewer for the references. We have developed our discussion based on these papers (lines 419-421; lines 517-519; lines 416-417 in tracked changes version of the manuscript / lines 392-394; lines 482-483; lines 389-390 in clean version of the manuscript) , except this of Canan et al, which concerned males. Indeed, following the comment of the reviewer 2, we have excluded males from our analyses.

Round 2

Reviewer 1 Report

Thank you for addressing my comments and suggestions - the paper has been greatly strengthened.

Reviewer 2 Report

In my opinion, the paper has been significantly improved.